# TrackingWorld: World-centric Monocular 3D Tracking of Almost All Pixels

**Jiahao Lu**[1]    **Weitao Xiong**[1,5]    **Jiacheng Deng**[2]    **Peng Li**[1]
**Tianyu Huang**[3]    **Zhiyang Dou**[4]    **Cheng Lin**[6]    **Sai-Kit Yeung**[1]    **Yuan Liu**[1†]

[1]HKUST    [2]USTC    [3]CUHK    [4]HKU    [5]XMU    [6]MUST

`https://github.com/IGL-HKUST/TrackingWorld`

## Abstract

Monocular 3D tracking aims to capture the long-term motion of pixels in 3D space from a single monocular video and has witnessed rapid progress in recent years. However, we argue that the existing monocular 3D tracking methods still fall short in separating the camera motion from foreground dynamic motion and cannot densely track newly emerging dynamic subjects in the videos. To address these two limitations, we propose TrackingWorld, a novel pipeline for dense 3D tracking of almost all pixels within a world-centric 3D coordinate system. First, we introduce a tracking upsampler that efficiently lifts the arbitrary sparse 2D tracks into dense 2D tracks. Then, to generalize the current tracking methods to newly emerging objects, we apply the upsampler to all frames and reduce the redundancy of 2D tracks by eliminating the tracks in overlapped regions. Finally, we present an efficient optimization-based framework to back-project dense 2D tracks into world-centric 3D trajectories by estimating the camera poses and the 3D coordinates of these 2D tracks. Extensive evaluations on both synthetic and real-world datasets demonstrate that our system achieves accurate and dense 3D tracking in a world-centric coordinate frame.

## 1  Introduction

Estimating long-term motion in dynamic videos remains a persistent challenge in computer vision [1, 2, 3, 4]. Fine-grained motion tracking is crucial for understanding object dynamics, modeling camera motion, and facilitating the generation of temporally and geometrically consistent videos [5, 6, 7].

In recent years, dense 2D pixel tracking [8, 9, 10, 1, 11, 12, 13, 14, 15, 16] has emerged as an active research topic, with notable advancements such as CoTrackers [17, 1], which employs transformers to iteratively update 2D tracks and has driven progress in 2D motion analysis. This development also motivates many recent works for 3D tracking. Early 3D tracking works like OmniMotion [2, 18] adopt optimization-based approaches to estimate 3D motion, while subsequent feedforward methods such as SpatialTracker [3] and DELTA [4] leverage extracted features to directly estimate the 3D tracking in a feedforward manner without per-sequence optimization. These 3D tracking methods demonstrate substantial potential for downstream applications, including detailed 3D motion analysis and high-fidelity novel view synthesis, highlighting the growing importance of monocular 3D tracking as a critical research frontier.

Upon analyzing all existing 3D tracking methods, we observe that these existing methods still suffer from two noticeable shortcomings. First, these methods [4, 3, 2] cannot distinguish the camera

---

†Corresponding authors

39th Conference on Neural Information Processing Systems (NeurIPS 2025).

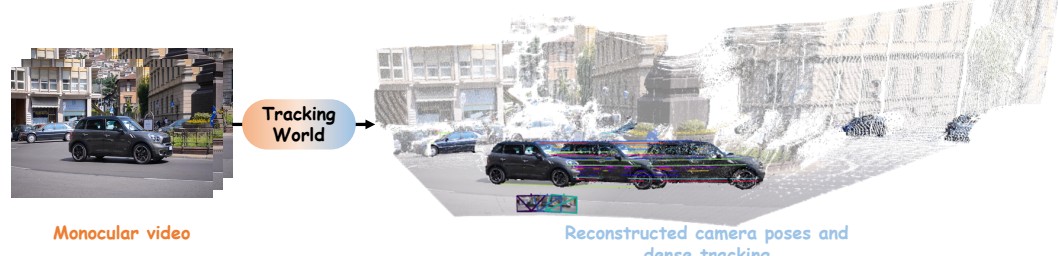

**Monocular video**

**Reconstructed camera poses and dense tracking**

Figure 1: **TrackingWorld** estimates world-centric dense tracking results from monocular videos. Our model can accurately estimate camera poses and achieve disentangled 3D track modeling of static and dynamic components, not just limited to one foreground dynamic object. We only visualize a subset of foreground dynamic point trajectories and apply a fading color to background static points.

motion and the dynamic object motion. All these methods assume a static camera and just model the 3D flow within the camera coordinate system. However, many downstream tasks like motion analysis or novel-view-synthesis require distinguishing camera motion from the dynamic object motion. Moreover, some recent works [19] also show that explicitly considering camera poses in motion estimation improves the 3D tracking quality. Only some very recent works [20, 21, 22] try to estimate the 3D tracks in the world-centric coordinate system and enable distinguishing camera motions from dynamic object motions. Estimating camera motion is still challenging for a monocular video containing dynamic objects because only static scenes provide cues for camera pose estimation.

The second shortcoming is that existing methods are mostly limited to tracking sparse pixels in the first frame of the video and cannot track all pixels in all frames (e.g., new objects emerging in the intermediate frames). Tracking all pixels brings a huge computational complexity to all tracking methods. Recent works like DELTA [4] propose to upsample the sparse tracking points with neural networks to produce dense 3D tracks. However, DELTA is still limited to tracking the first frame of the video, and how to estimate the dense 3D tracks for all pixels of all frames still remains an unexplored problem.

In this paper, we propose **TrackingWorld**, a 3D tracking method that enables dense 3D tracking of almost all pixels of all frames from a monocular video within a world-centric coordinate system. "almost all" means we filter some noisy and outlier tracks to ensure robustness and accuracy. Specifically, TrackingWorld takes a monocular video and the monocular estimation from foundation models as input, including sparse tracks [4, 1], depth maps [23], and coarse foreground dynamic masks [24]. Then, TrackingWorld produces high quality dense 3D tracks for almost all pixels of the monocular video and the camera poses for every frame. TrackingWorld addresses the above shortcomings with the following strategies.

First, to enable the dense tracking of almost all pixels, we utilize the track upsampler of DELTA [4] and track every frame iteratively. We find that the tracking upsampler module of DELTA [4] is applicable to arbitrary 2D tracks, which are utilized by TrackingWorld to upsample the input sparse 2D tracks to dense 2D tracks. Then, we not only track the pixels of the first frame but also repeat it on all subsequent frames. To reduce computational complexity, we observe that many regions of subsequent frames have already been seen in the first or previous frames. Therefore, we delete the redundant tracks corresponding to these overlapping regions.

Second, to accurately separate the camera motion from the dynamic object motion, we estimate the 3D tracks and the camera poses from the upsampled dense 2D tracks and the input estimated depth maps. A key challenge lies in the inaccuracy of the estimated dynamic masks, which often fail to capture dynamic background objects. This limitation leads to suboptimal bundle adjustment interfered by dynamic background objects, ultimately compromising the accuracy of both camera pose estimation and object motion tracking. Thus, we treat all points in the initial static regions as potentially dynamic but impose an as-static-as-possible constraint for the camera pose estimation, which effectively helps us rule out the dynamic background points for an accurate camera pose estimation. Finally, we utilize the estimated camera poses along with the depth maps to convert all the 2D tracks into 3D tracks in the world coordinate.

To comprehensively evaluate whether our proposed method can effectively achieve dense 3D tracking of almost all pixels across all frames within a world-centric coordinate system, we conduct evaluations

from multiple perspectives: 1. Camera pose estimation accuracy; 2. Depth accuracy of the dense 3D tracks; 3. Sparse 3D tracking performance; 4. Accuracy of the dense 2D tracking results. Our empirical analysis demonstrates that the proposed method yields superior performance across all metrics, confirming its effectiveness in establishing accurate and consistent 3D tracks over time.

## 2 Related Work

### 2.1 2D Point Tracking

The task of tracking arbitrary points [8, 9, 10, 1, 11, 12, 13, 14, 15, 16] across video frames is first introduced by PIPs [8], which leverages deep learning to tackle point tracking based on optical flow. Built upon RAFT [25], PIPs computes inter-frame correlation maps and uses a decoder to iteratively refine tracking results. TAP-Vid [9] further improves the problem formulation, introducing three standardized benchmarks along with TAP-Net, a dedicated model for point tracking. TAPIR [10] advances performance by combining a matching stage with a refinement stage, enhancing tracking accuracy. CoTrackers [17, 1] observe that strong correlations exist across different point trajectories, and exploit this insight by training on unrolled sequences over long videos, which significantly improve long-term tracking performance. Drawing inspiration from DETR [26], TAPTR [12] proposes an end-to-end transformer-based architecture, where each point is represented as a query token in the decoder, enabling direct modeling of point dynamics. LocoTrack [11] extends traditional 2D correlation features to 4D correlation volumes and introduces a lightweight correlation encoder, achieving better efficiency while preserving accuracy.

### 2.2 3D Point Tracking

While previous works have primarily focused on 2D point tracking, recent research has increasingly focused on 3D point tracking [2, 18, 3, 4, 27, 28, 29, 30, 31, 22, 20, 21]. Early 3D tracking methods, such as OmniMotion [2], adopt optimization-based approaches to estimate 3D motion. Subsequent work like OmniTrackFast [18] aims to reduce the optimization time and enhance robustness. More recently, increasing attention has shifted toward feedforward-based methods. For example, Spatial-Tracker [3] represents points in a (u, v, d) coordinate system, combining image-plane coordinates with depth information. It incorporates depth priors and uses a triplane representation to enable effective 3D tracking. Building upon this idea, DELTA [4] also adopts the UVD coordinate system, but takes a different approach by decoupling appearance and depth correlations. DELTA introduces a coarse-to-fine trajectory estimation strategy, allowing for efficient dense tracking across the entire frame rather than being limited to a sparse set of locations. In contrast to the aforementioned methods that focus on UVD (2.5D) representations, several concurrent works have recently explored 3D tracking in a world-centric coordinate system. St4RTrack [20] adopts a DUSt3R [32]-like framework to establish pairwise correspondences, but this approach may suffer from drift during long-term tracking. TAPIP3D [21] primarily focuses on sparse tracking and is inherently unable to recover camera motion. In comparison, our method introduces a comprehensive pipeline for dense 3D tracking that can robustly capture newly emerging objects within a world-centric coordinate system.

### 2.3 4D Reconstruction

4D reconstruction [33, 34, 35, 36, 19, 37, 38, 39, 40, 24] aims to recover both camera motion and object motion within a scene. The problem of non-rigid structure from motion is highly ill-posed. To overcome this limitation, a variety of approaches have been proposed. RobustCVD [33] refines depth estimation using 3D geometric constraints, while CasualSAM [34] finetunes a depth network guided by predicted motion masks. MegaSaM [35] integrates monocular depth priors and motion probability maps into a differentiable SLAM paradigm. Inspired by DUSt3R [32], several data-driven methods such as MonST3R [37], Align3R [38], and Cut3r [40] adopt 3D point cloud representations to enable full 4D reconstruction. In addition, Uni4D [24], a multi-stage optimization framework, leverages multiple pretrained models to improve reconstruction in dynamic scenes. Its core contribution lies in the use of foundation models to achieve effective separation of static and dynamic elements within the scene. Our method also adopts an optimization-based framework to decouple camera motion and object motion. However, unlike prior works that primarily focus on 4D reconstruction, our approach targets a higher-level task—dense tracking of every pixel—which enables fine-grained correspondence estimation across time. By focusing on dense pixel-level tracking, our method

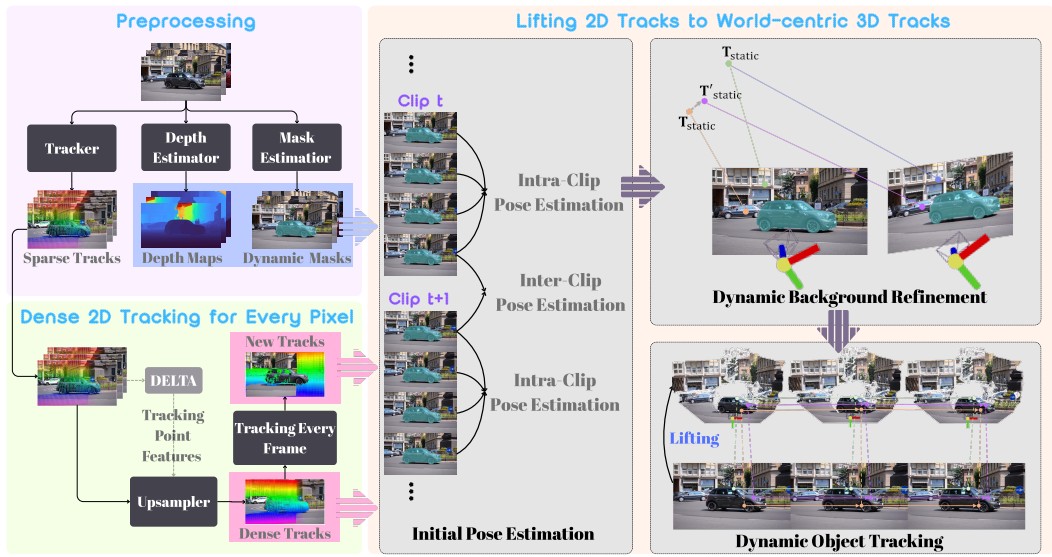

Figure 2: **Overview**. Given a video sequence, TrackingWorld first generates dense 2D tracking results that are capable of capturing newly emerging objects in the scene. These 2D trajectories are then fed into an optimization-based framework to transform them into a world-centric 3D space. Specifically, we begin by estimating the initial camera poses for each frame at the clip level. We then perform dynamic background refinement to exclude potentially dynamic regions and refine the camera poses. Based on the optimized poses, we finally reconstruct the trajectories of all dynamic regions.

provides a more detailed and temporally consistent understanding of dynamic scenes, making it well-suited for applications such as motion analysis, scene understanding, and video editing [5, 6, 7].

## 3 Method

### 3.1 Overview

Given a video consisting of $T$ frames $\{\mathbf{I}_t \in \mathbb{R}^{\mathrm{H} \times \mathrm{W} \times 3} \mid t = 1, \ldots, T\}$, the goal of TrackingWorld is to estimate the corresponding dense 3D trajectories (3D tracks) $\{\mathbf{T}_t \in \mathbb{R}^{\mathrm{M}_t \times 3} \mid t = 1, \ldots, T\}$ of almost all pixels, where $\mathrm{M}_t$ denotes the number of tracked points at timestep $t$, along with the camera poses $\{\pi_t \in \mathbb{R}^{3 \times 4} \mid t = 1, ..., T\}$. Our proposed **TrackingWorld** framework, illustrated in Fig. 2, achieves this through two main components: first, generating dense 2D tracking results that are capable of following nearly every object in the scene; second, back-projecting these dense 2D tracking results into a world-centric 3D space.

**Preprocessing with vision foundation models.** For the monocular video, we first preprocess it with a 2D tracking model, a foreground dynamic mask estimation module, and a monocular depth estimation module to get a set of 2D tracks, dynamic masks, and depth maps for all frames. For the 2D tracking model, we choose the CoTrackerV3 [1] or the 2D tracking part of DELTA [4]. For the dynamic mask estimation method, we follow Uni4D [24] to apply VLM [41] and Grounding-SAM [42, 43] to segment out foreground dynamic objects. Alternatively, we could also choose the SegmentAnyMotion [44] to get dynamic masks. For the depth estimation, we choose UniDepth [23]. Note that all these predictions are not required to be accurate, and we may also adopt other foundation models for this purpose.

### 3.2 Dense 2D Tracking for Every Pixel

In this section, our target is to achieve dense 2D tracking of almost any pixels in the video. We achieve this through two modules: First, we lift the input sparse 2D tracks for a frame to dense 2D tracks; Second, we repeat tracking on every frame and eliminate the overlapped redundant 2D tracks.

**Sparse to dense tracks.** Given the sparse 2D tracks $\mathbf{P}_{\text{sparse}} \in \mathbb{R}^{(\frac{H}{s} \times \frac{W}{s}) \times T \times 2}$ for a specific frame, this module aims to lift the sparse 2D tracks to dense 2D tracks $\mathbf{P}_{\text{dense}} \in \mathbb{R}^{(H \times W) \times T \times 2}$. $s$ means the downsampled factor. We achieve this by utilizing the upsampler module of DELTA [4]. The upsampler module takes the sparse tracks $\mathbf{P}_{\text{sparse}}$ and features defined on the sparse 2D tracks $\mathbf{F}_{\text{sparse}} \in \mathbb{R}^{\frac{H}{s} \times \frac{W}{s} \times T \times C}$, where $C$ is the feature dimension, as inputs and predicts a weight matrix $\mathbf{W} \in \mathbb{R}^{(\frac{H}{s} \times \frac{W}{s}) \times (H \times W)}$. Then, the upsampled dense 2D tracks are

$$\mathbf{P}_{\text{dense}} = \mathbf{W}^T \mathbf{P}_{\text{sparse}}, \tag{1}$$

where $\mathbf{W}$ actually only correlates a dense track in $\mathbf{P}_{\text{dense}}$ with its neighboring 2D tracks in $\mathbf{P}_{\text{sparse}}$. We find that this upsampler module is not only compatible with DELTA's 2D tracks but also generalizes to arbitrary 2D tracks, so we adopt it here to upsample the arbitrary input sparse 2D tracks into dense 2D tracks for a specific frame.

**Tracking every frame.** Based on the above upsampler, we further enable tracking of almost all pixels of all frames. To achieve this, we conduct 2D tracking and the sparse-to-dense upsampling on all frames in the video. However, this leads to a large redundancy on the tracking points because most regions are already seen in the previous frames, while only a few regions are new. To avoid wasting computation on these redundant 2D tracks in the subsequent computation, if a pixel resides near the tracking trajectory of arbitrary visible previous 2D tracks, then we discard the pixel. More details can be found in A.6 of the supplementary material.

### 3.3 Lifting 2D Tracks to World-centric 3D Tracks

In this module, we will estimate the camera poses of all frames and lift the dense 2D tracks estimated by the previous section to 3D tracks in the world-centric coordinate system. We achieve this through the following three steps: First, we utilize the input estimated coarse dynamic masks and estimate the camera poses using only the coarse static regions. However, the dynamic masks are usually not accurate enough, and some dynamic objects in the background still remain. Second, we utilize an as-static-as-possible constraint to further improve the camera pose estimation and find out the dynamic objects in the background. Finally, we transform all 2D tracks within the dynamic regions into 3D tracks in the world-centric coordinate system.

**Initial camera pose estimation.** In this step, we want to estimate per-frame camera poses $\{\pi_t \in \text{SE}(3)\}$ from the 2D tracks on static regions and the estimated depth maps. We first utilize the input dynamic foreground masks to select 2D tracks $\mathbf{P}_{\text{static}} \in \mathbb{R}^{N_{\text{static}} \times T \times 2}$ on these static regions. Then, for each static 2D track, we unproject its location at timestep $t_1$ into the 3D space using the monocular depth map $\mathbf{D}_{\text{static}} \in \mathbb{R}^{N_{\text{static}} \times T \times 1}$. The resulting 3D points are subsequently reprojected into the image plane at timestep $t_2$ using the camera poses. Then, we define the projection loss to optimize the camera poses

$$\mathcal{L}_{\text{proj}} = \sum_i^{N_{\text{inliers}}} \sum_{t_1}^T \sum_{t_2}^T \|\pi_{t_2} \pi_{t_1}^{-1}(\mathbf{P}_{\text{static}}(i, t_1), \mathbf{D}_{\text{static}}(i, t_1)) - \mathbf{P}_{\text{static}}(i, t_2)\|_2^2, \tag{2}$$

where $\pi_t(\cdot)$ means project with the camera pose on timestep $t$, and $\mathbf{P}_{\text{static}}(i, t) \in \mathbb{R}^2$ means the position of the $i$-th static 2D track on timestep $t$, $\mathbf{D}_{\text{static}}(i, t)$ means the depth value for the $i$-th static point on time step $t$, and $N_{\text{inliers}}$ denotes the number of static 2D tracks whose projection errors fall within the threshold $\tau$.

To further improve the computational efficiency for camera pose estimation, we first divide the entire video into $C$ clips and estimate camera poses within each clip in parallel. After estimating camera poses within each clip, we estimate the pose between clips to merge the camera poses together.

**Dynamic background refinement.** The foreground dynamic object masks are usually not accurate enough, so some dynamic objects in the background still exist in the assumed "static" regions and prevent us from accurately estimating camera poses. Thus, we further refine the camera pose estimation by treating these static regions as dynamic and introducing an as-static-as-possible constraint.

Specifically, each static 2D track corresponds to a unique 3D point in the world-centric coordinate system, denoted as $\mathbf{T}_{\text{static}} \in \mathbb{R}^{N_{\text{static}} \times 3}$. We initialize $\mathbf{T}_{\text{static}}$ by back-projecting the static 2D tracks

using the depth estimated by UniDepth and the camera poses obtained from the previous stage: Initial camera pose estimation. Notably, for each 2D track $\in \mathbb{R}^{T \times 2}$, we only back-project the visible timesteps and take the average of the resulting 3D points. To better model the potentially dynamic regions that are not accurately segmented, we introduce an additional object motion term $\mathbf{O}_{\text{static}} \in \mathbb{R}^{N_{\text{static}} \times T \times 3}$, which captures residual object motions over time. With this term, the time-dependent world-centric static tracking becomes

$$\mathbf{T}'_{\text{static}}(i,t) = \mathbf{T}_{\text{static}}(i) + \mathbf{O}_{\text{static}}(i,t), \tag{3}$$

where $\mathbf{T}'_{\text{static}}(i,t) \in \mathbb{R}^3$ means the 3D coordinate of the $i$-th static point at timestep $t$ and $\mathbf{O}_{\text{static}}(i,t)$ is the corresponding 3 dimensional offset. We then jointly optimize the camera poses $\pi_t$ and the static 3D coordinates $\mathbf{T}'_{\text{static}}$ using a bundle adjustment loss:

$$\mathcal{L}_{\text{ba}} = \sum_{i=1}^{N_{\text{static}}} \sum_{t=1}^{T} \left\| \pi_t(\mathbf{T}'_{\text{static}}(i,t)) - \mathbf{P}_{\text{static}}(i,t) \right\|_2^2, \tag{4}$$

where $\mathbf{P}_{\text{static}}(i,t)$ is the observed 2D projection of the $i$-th track at timestep $t$. In addition to the bundle adjustment loss, we also compute a depth consistency loss $\mathcal{L}_{\text{dc}}$ to enforce the consistency between the projected depth maps from $T'_{\text{static}}$ and the estimated monocular depth maps, as introduced in the supplementary material. To ensure that residual motion remains minimal for genuinely static regions, we regularize the offset $\mathbf{O}_{\text{static}}$ with an as-static-as-possible constraint

$$\mathcal{L}_{\text{asap}} = \sum_{i,t} \left\| \mathbf{O}_{\text{static}}(i,t) \right\|_1, \tag{5}$$

where we minimize the L1 norms of offsets to make all points as static as possible. This $\mathcal{L}_{\text{asap}}$ enables the accurate camera estimation and also models the dynamics of background objects.

**Dynamic object tracking.** In this step, our target is to lift the 2D tracks of dynamic regions to 3D tracks. We also include the dynamic background points with $\|\mathbf{O}_{\text{static}}(i,\cdot)\|_2 \geq \varepsilon$ here as the dynamic 3D tracks. For these dynamic 3D tracks, we directly represent their 3D coordinates by $\mathbf{T}_{\text{dynamic}} \in \mathbb{R}^{N_{\text{dynamic}} \times T \times 3}$. Similar to the 3D static tracks, we initialize the dynamic 3D tracks by back-projecting them using the depths predicted by UniDepth and the camera poses refined in the second stage. Based on $\mathbf{T}_{\text{dynamic}}$, we also compute the projection loss in Eq. 4, the depth consistency loss $\mathcal{L}_{\text{dc}}$, as-rigid-as-possible loss $\mathcal{L}_{\text{arap}}$ [45, 24], and a temporal smoothness loss $\mathcal{L}_{\text{ts}}$ [24]. All the details of these loss terms are included in the supplementary material. The final outputs are the dynamic 3D tracks $\mathbf{T}_{\text{dynamic}}$, static 3D tracks $\mathbf{T}'_{\text{static}}$, and the camera poses $\pi_t$.

**Discussion**. The tracking module TrackingWorld differs from previous 3D tracking methods, DELTA [4] and SpatialTracker [3] by explicitly estimating the camera poses, which enables the estimation of 3D tracks in the world-centric coordinate system. The explicit separation between camera motion and object motion also improves the quality of 3D tracking because of the better decomposition, as demonstrated by experimental results in Tab. 3. In comparison with the existing dynamic video camera pose estimation methods, like Uni4D [24], we do not just assume a single dynamic foreground object but also model the background object motion in the camera pose estimation for a better performance. Instead of simply discarding these dynamic background objects, we also track their 3D points in the world-centric coordinate system, enabling tracking almost all pixels.

## 4 Experiment

### 4.1 Implementation details

All experiments are conducted on an RTX 4090 GPU. We use CoTrackerV3 [1] and DELTA [4] to obtain dense tracking results, and adopt UniDepth [24] as the depth prior. The entire framework takes ∼20 minutes to produce dense world-centric 3D tracking results for a 30-frame video. All baseline methods are run on the datasets using their official implementations and default hyperparameters. More details about hyperparameters can be found in the supplementary materials.

### 4.2 Quantitative comparisons

To demonstrate the capability of our method in dense 3D tracking within a world-centric coordinate system, we evaluate the following performance: 1. Camera pose estimation accuracy; 2. Depth accuracy of dense 3D tracks; 3. Sparse 3D tracking performance; 4. Dense 2D tracking performance.

### 4.2.1 Camera pose estimation results

**Benchmarks and metrics.** We evaluate camera pose estimation performance on three dynamic datasets: Sintel [46], Bonn [47], and TUM-D [48]. For all three datasets, we adopt the same settings as MonST3R [37]. Following [49, 50, 51], we report three ATE ↓ (Absolute Trajectory Error), RTE ↓ (Relative Translation Error), and RRE ↓ (Relative Rotation Error). ATE measures the deviation between estimated and ground truth trajectories after alignment. RTE and RRE evaluate the average local translation and rotation errors over consecutive pose pairs, respectively.

**Comparison with existing methods.** Tab. 1 presents the quantitative comparison between our method and existing approaches. To recover the camera pose, we first obtain dense tracking results, followed by optimization process that refines the camera pose and world-centric dense tracking. As shown in the table, regardless of whether the dense tracking is derived from DELTA [4] or CoTrackerV3 [1], our method consistently achieves more accurate pose estimation than previous approaches across all three datasets.

| Category | Method | Sintel ATE ↓ | Sintel RTE↓ | Sintel RRE↓ | Bonn ATE ↓ | Bonn RTE↓ | Bonn RRE↓ | TUM-D ATE ↓ | TUM-D RTE↓ | TUM-D RRE↓ |
|---|---|---|---|---|---|---|---|---|---|---|
| Pose only | DROID-SLAM‡ [52] | 0.175 | 0.084 | 1.912 | / | / | / | / | / | / |
| | DPVO‡ [51] | 0.115 | 0.072 | 1.975 | / | / | / | / | / | / |
| | COLMAP [53] | 0.559 | 0.325 | 7.302 | / | / | / | 0.076 | 0.059 | 7.689 |
| Joint depth & pose | Robust-CVD [33] | 0.360 | 0.154 | 3.443 | / | / | / | 0.153 | 0.026 | 3.528 |
| | DUSt3R [32] | 0.601 | 0.214 | 11.43 | 0.046 | 0.014 | 1.836 | 0.083 | 0.017 | 3.567 |
| | MonST3R [37] | 0.111 | 0.044 | 0.780 | 0.029 | 0.007 | 0.612 | 0.063 | 0.009 | 1.217 |
| | Align3R [38] (Depth Pro [54]) | 0.128 | 0.042 | 0.432 | 0.023 | 0.007 | 0.620 | 0.027 | 0.018 | 0.446 |
| | Uni4D* [24] | 0.116 | 0.046 | 0.603 | 0.017 | 0.006 | 0.561 | 0.039 | 0.007 | 0.434 |
| | Ours (CoTrackerV3 [1]) | 0.103 | 0.039 | 0.439 | 0.016 | **0.005** | **0.561** | **0.014** | **0.005** | 0.338 |
| | Ours (DELTA [4]) | **0.088** | **0.035** | **0.410** | **0.016** | 0.005 | 0.564 | 0.016 | 0.005 | **0.333** |

Table 1: **Camera pose estimation results.** We evaluate our model on three datasets: Sintel, Bonn, and TUM-D. **Best** results are highlighted. ‡ means using ground truth camera intrinsics as input. * means reproduced by 2D tracks from DELTA, the same as "Ours(DELTA)".

### 4.2.2 Depth accuracy of the dense 3D tracks

**Benchmarks and metrics.** Since our method does not aim to optimize 2D tracking accuracy directly, but rather focuses on how to transform 2D tracking into dense world-centric tracking, we evaluates the accuracy of the camera-centric depth for each tracked point. Specifically, we compare the predicted depth with the ground-truth depth only for tracked points that lie within the image bounds. As multiple tracked points may track the same pixel, we retain the one with the smaller depth value for evaluation, assuming it more likely corresponds to the visible surface. Similar to the camera pose benchmark, we evaluate on the same datasets and under identical settings: Sintel, Bonn, and TUM-D. Following prior works [55, 37], we align the estimated dense tracking depth with the ground truth using a single scale and shift before computing the evaluation metrics. We primarily report two metrics: Abs Rel ↓ (absolute relative error) and the percentage of inlier points with $\delta < 1.25$ ↑.

**Comparison with existing methods.** Tab. 2 reports the results of dense tracking depth estimation. Thanks to our optimization-based bundle adjustment, which enforces strong 3D geometric consistency, the estimated tracking depth is significantly improved across all datasets.

| Method | Depth Prior | Sintel Abs Rel ↓ | Sintel $\delta < 1.25$ ↑ | Bonn Abs Rel ↓ | Bonn $\delta < 1.25$ ↑ | TUM-D Abs Rel ↓ | TUM-D $\delta < 1.25$ ↑ |
|---|---|---|---|---|---|---|---|
| DELTA [4] | ZoeDepth [56] | 0.814 | 46.1 | 0.168 | 88.5 | 0.239 | 70.5 |
| DELTA [4] | Depth Pro [54] | 0.813 | 50.7 | 0.160 | 90.6 | 0.222 | 78.4 |
| DELTA [4] | Unidepth [23] | 0.636 | 63.1 | 0.153 | 90.5 | 0.178 | 85.6 |
| Ours (CoTrackerV3 [1]) | Unidepth [23] | 0.219 | 73.1 | **0.054** | 97.2 | 0.089 | 91.5 |
| Ours (DELTA [4]) | Unidepth [23] | **0.218** | **73.3** | 0.058 | **97.3** | **0.084** | **92.3** |

Table 2: **Depth accuracy of the dense 3D tracks. Best** results are highlighted.

| Category | Method | ADT | | | PStudio | | |
|---|---|---|---|---|---|---|---|
| | | AJ↑ | $APD_{3D}$↑ | OA↑ | AJ↑ | $APD_{3D}$↑ | OA↑ |
| Feed. | CoTrackerV3 [1]+Uni [23] | 13.6 | 21.3 | 88.5 | 14.1 | 22.8 | **87.7** |
| | SpatialTracker [3] | 14.3 | 22.3 | **91.5** | 13.8 | 23.7 | 79.5 |
| | DELTA [4] | 15.3 | 22.9 | 90.1 | 15.1 | 24.6 | 75.7 |
| Optim. | OmniTrackFast [18] | 8.6 | 18.2 | 63.9 | 6.4 | 12.2 | 81.8 |
| | Ours (CoTrackerV3 [1]) | 22.5 | 31.5 | 88.5 | 14.2 | 24.0 | **87.7** |
| | Ours (DELTA [4]) | **23.4** | **32.2** | 90.1 | **15.1** | **25.6** | 75.7 |

| Method | CVO-Clean | | CVO-Final | |
|---|---|---|---|---|
| | EPE↓ | IoU↑ | EPE↓ | IoU↑ |
| RAFT [25] | 2.48 | 57.6 | 2.63 | 56.7 |
| CoTracker [17] | 1.51 | 75.5 | 1.52 | 75.3 |
| SpatialTracker [3] | 1.84 | 68.5 | 1.88 | 68.1 |
| DOT-3D [59] | 1.33 | 79.0 | 1.38 | 78.8 |
| DELTA [4] | **1.14** | 78.9 | 1.39 | 78.2 |
| CoTrackerV3 [1] + Up | 1.24 | **80.9** | **1.35** | **80.6** |

Table 3: **Sparse 3D tracking results.** "Feed." means feedforward methods while "Optim" means optimization-b ased method.

Table 4: **Long-range optical flow results.**

### 4.2.3 Sparse 3D tracking results

**Benchmarks and Metrics.**  To evaluate the performance of 3D sparse tracks, we conduct experiments on two datasets, ADT [57] with moving cameras, and PStudio [58] with static cameras. For each dataset, the video subsets for evaluation are selected at fixed intervals: for ADT, we sample one video every 100 videos, and for PStudio, one video every 20 videos. The sparse 3D tracking result are evaluated in camera coordinates. As for evaluation metrics, we adopt Average Jaccard (AJ), which jointly evaluates the accuracy of both spatial position and occlusion estimation, serving as a comprehensive indicator of tracking quality; $APD_{3D}$ ($< \delta_{avg}$) which measures the average percentage of tracked points whose errors fall within a given threshold $\delta$, reflecting geometric accuracy; Occlusion Accuracy (OA) which evaluates the precision of occlusion state prediction across frames.

**Comparison with existing methods.**  Since our method primarily focuses on dense tracking, we maintain the optimization of dense tracking results even when evaluating sparse tracking performance. To this end, we sample evaluation points from the optimized dense tracking set. As shown in Tab. 3, our method achieves higher 3D geometric consistency in tracking. For scenes with camera motion (ADT), the explicit separation between camera motion and object motion leads to significant improvements in both AJ and $APD_{3D}$. In contrast, for scenes with static cameras (PStudio), the benefits from geometric optimization are relatively limited, resulting in smaller performance gains. It is worth noting that OA mainly evaluates the visibility accuracy of tracking points. Since we directly adopt the visibility maps predicted by DELTA/CoTrackerV3, the OA scores remain consistent with those of DELTA/CoTrackerV3.

### 4.2.4 Accuracy of dense 2D tracks

**Benchmarks and Metrics.**  We evaluate the dense 2D tracking performance on the CVO [60] test set, which consists of two subsets: CVO-Clean and CVO-Final, with the latter incorporating motion blur. Each subset contains approximately 500 videos with 7 frames. For evaluation, we adopt the following metrics: the end-point error (EPE) between the predicted and ground-truth optical flows for all points, and the intersection-over-union (IoU) between the predicted and ground-truth occluded regions in visible masks.

**Comparison with existing methods.**  To verify the accuracy of the 2D dense tracks generated by the upsampler module (Up) introduced in Sec. 3.2, we conduct additional long-range optical flow experiments, as shown in Tab. 4. The results demonstrate that the upsampler module generalizes well to other 2D trackers, such as CoTrackerV3, achieving comparable performance with DELTA.

### 4.3 Qualitative results

Fig. 3 qualitatively visualizes the world-centric dense tracking results produced by our method on the DAVIS [61] dataset. For each video sequence, the second row displays 3D tracking results on temporally spaced keyframes, making the changes in object trajectories more perceptible while avoiding visual clutter. The third row presents continuous 3D tracks across all frames, offering a comprehensive view of motion consistency and trajectory completeness. As discussed in Sec. 3.3, by separating dynamic and static elements, we can generate stable tracking results for both the static background and dynamic objects.

### 4.4 Ablation study

**Ablation study on the different components.**  As shown in Tab. 5, we conduct ablation studies to validate our major design choices. Specifically, the different configurations are as follows: 1) without

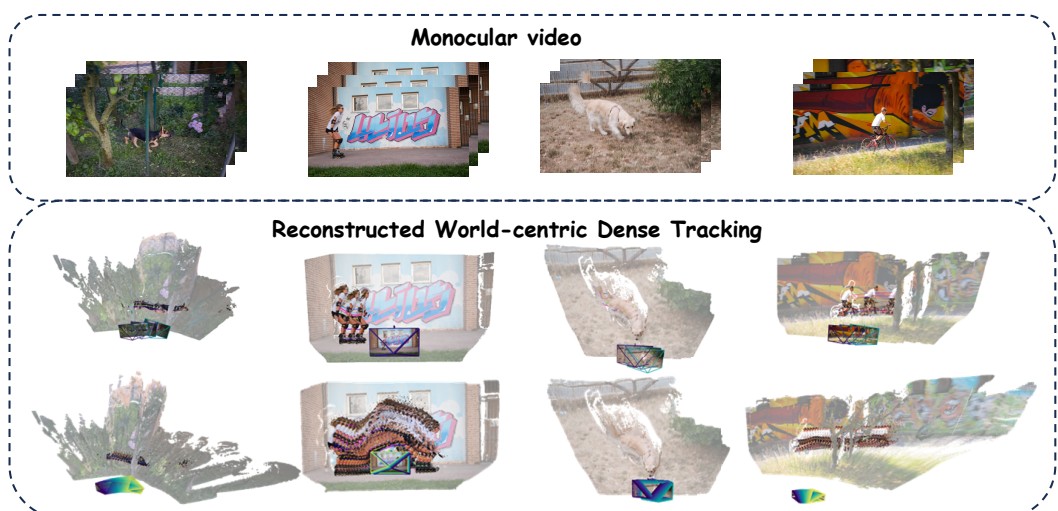

Figure 3: **Qualitative results on DAVIS dataset.** Our method can output both reliable camera trajectories and world centric dense tracking. The second row visualizes 3D tracking results on temporally spaced keyframes, while the third row shows complete tracks across continuous frames.

| Setting | Sintel | | | | |
|---|---|---|---|---|---|
| | ATE ↓ | RTE↓ | RRE↓ | Abs Rel ↓ | $\delta < 1.25$ ↑ |
| w/o T.E.F | 0.171 | 0.047 | 0.748 | / | / |
| w/o pose-init. | 0.659 | 0.153 | 1.382 | 0.230 | 72.4 |
| w/o D.O.T | 0.088 | 0.035 | 0.410 | 0.468 | 73.0 |
| w/o $N_{\text{inliers}}$ | 0.089 | 0.035 | 0.414 | 0.220 | 72.9 |
| w/o $\mathbf{O}_{\text{static}}$ | 0.092 | 0.036 | 0.459 | 0.224 | 72.6 |
| w/o $\mathcal{L}_{\text{dc}}$ | 0.093 | 0.036 | 0.441 | 0.234 | 71.2 |
| Full | **0.088** | **0.035** | **0.410** | **0.218** | **73.3** |

Table 5: **Ablation study on Sintel dataset.**

Figure 4: **Effectiveness of $\mathbf{O}_{\text{static}}$.** Key regions are highlighted in red.

tracking every frame (w/o T.E.F): In this setting, we only track from the first frame, which leads to the loss of many critical cues for pose estimation, thereby resulting in a significant performance drop. 2) without initial camera pose estimation (w/o pose-init.): We observe that under this setting, it becomes difficult to jointly optimize both camera poses and 3D tracks effectively — a good initialization of the camera poses is necessary to achieve satisfactory results. 3) without dynamic object tracking (w/o D.O.T): In this setting, the depths of dynamic tracks are directly obtained from UniDepth predictions without further refinement. As shown in the table, optimizing dynamic tracks is crucial for achieving better performance. 4) without selecting the inliers whose reprojection errors are within a threshold $\tau$ (w/o $N_{\text{inliers}}$): By filtering all static points and optimizing with nearly static points, we can effectively reduce the influence of outlier trajectories and obtain more accurate camera poses. 5) without the object motion term $\mathbf{O}_{\text{static}}$ (w/o $\mathbf{O}_{\text{static}}$): We do not consider the dynamic objects in the assumed "static" background and directly optimize the camera poses with all "static" background. We show the projected background "static" points (in green and red dots) in Fig. 4. As we can see, the "apple" (in the red dots) is considered a background static region but is actually dynamic. Without modeling dynamic points in the background, the points of the apple are incorrectly projected onto incorrect regions. 6) without the depth consistency loss (w/o $\mathcal{L}_{\text{dc}}$): $\mathcal{L}_{\text{dc}}$ can enforce the consistency between the projected depths and the estimated monocular depths, which helps suppress abnormal depth estimations to some extent.

**Ablation on different depth estimation models.** We conducted an ablation study using three commonly used monocular depth estimation models: ZoeDepth [56], Depth Pro [54], and UniDepth [23]. For all experiments, we fixed the tracking component to DELTA [4] and evaluated both the camera pose estimation accuracy and the depth accuracy of the dense 3D tracks on the Sintel dataset. As shown in Tab. 6, our method consistently improves over raw depth predictions across all depth models, especially in downstream tasks such as camera pose estimation. This demonstrates that our pipeline is robust to different depth estimation backbones.

| Method | ATE ↓ | RTE ↓ | RPE ↓ | Abs Rel ↓ | $\delta < 1.25$ ↑ |
|---|---|---|---|---|---|
| ZoeDepth | / | / | / | 0.814 | 46.1 |
| Depth Pro | / | / | / | 0.813 | 50.7 |
| UniDepth | / | / | / | 0.636 | 63.1 |
| Ours (ZoeDepth) | 0.093 | 0.038 | 0.418 | 0.236 | 72.1 |
| Ours (Depth Pro) | 0.101 | 0.036 | 0.434 | 0.228 | 72.6 |
| Ours (UniDepth) | **0.088** | **0.035** | **0.410** | **0.218** | **73.3** |

Table 6: Ablation study on different depth estimation models.

**Ablation on dynamic mask segmentators.** As shown in Tab. 7, we also evaluate different sources of dynamic mask segmentations and observe comparable performance, further demonstrating the robustness of our pipeline.

| Method | ATE ↓ | RTE ↓ | RPE ↓ | Abs Rel ↓ | $\delta < 1.25$ ↑ |
|---|---|---|---|---|---|
| Ours + VLM + GroundingSAM | **0.088** | **0.035** | 0.410 | **0.218** | 73.3 |
| Ours + Segment Any Motion | 0.093 | 0.041 | **0.379** | 0.224 | **73.3** |

Table 7: Ablation study on different dynamic mask segmentators.

**Necessity of the 2D upsampler module.** The 2D upsampler is crucial for achieving efficient dense tracking. Directly predicting dense 2D correspondences (e.g., using CoTrackerV3 [1]) is computationally expensive and memory-intensive, with no clear accuracy gain. To validate this, we compare CoTrackerV3 with and without our upsampler on the CVO-Clean dataset (7-frame sequences). As shown in Tab. 8, the upsampler improves both accuracy (lower EPE, higher IoU) and drastically reduces runtime (approximately $12\times$ speed-up). This supports our design choice.

| Method | EPE ↓ | IoU ↑ | Avg. Time (min) ↓ |
|---|---|---|---|
| CoTrackerV3 | 1.45 | 76.8 | 3.00 |
| CoTrackerV3 + Up | **1.24** | **80.9** | **0.25** |

Table 8: Ablation on the 2D upsampler module.

## 5 Conclusion

In this paper, we propose TrackingWorld, a novel method for dense 3D tracking of almost all pixels of all frames from a monocular video within a world-centric coordinate system. The key idea of TrackingWorld is to explicitly disentangle camera motion from foreground dynamic motion while densely tracking newly emerging objects. We first introduce a tracking upsampler to densify sparse 2D tracks and apply it to capture newly emerging objects. Finally, we design an efficient optimization-based framework to lift dense 2D tracks into consistent 3D world-centric trajectories. Extensive evaluations across multiple dimensions demonstrate the effectiveness of our system.

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
