# OpenReview forum: "TrackingWorld: World-centric Monocular 3D Tracking of Almost All Pixels"
_NeurIPS.cc/2025/Conference — NeurIPS 2025 poster_

### Official Review · Reviewer_yGCi · 2025-06-30

**Clarity:** 3
**Significance:** 3
**Originality:** 2
**Rating:** 4
**Confidence:** 5

**Summary:**

This paper proposes TrackingWorld, a novel pipeline for dense world-coordinate 3D tracking that estimates world-centric trajectories of almost all pixels from monocular video. The method introduces two key contributions:
* All-frame Dense 2D Tracking: The approach leverages sparse 2D tracking [1] and the upsampling layer from DELTA [2] to enable tracking of all pixels across all frames, including newly emerging objects, while eliminating redundant tracks.

* Optimization-based Camera Pose Estimation and World-centric 3D Tracking: A three-phase optimization strategy is proposed:
  * (1) Initial camera pose estimation using monocular depths and static 2D tracking.
  * (2) Joint refinement of camera poses and static 3D tracks.
  * (3) Estimation of dynamic 3D tracks.

The proposed method is evaluated on several benchmarks, including Sintel, Bonn, TUM-D, ADT, and PStudio, and demonstrates improvements over prior methods in pose estimation accuracy, 3D tracking quality, and depth consistency.

**Questions:**

**Questions**:
* L188–190: How are the camera poses from individual clips combined into a unified, globally consistent pose sequence for the entire video?
* The current approach uses dense 2D tracks and monocular depth for the optimization step. Could the method alternatively leverage dense 3D tracks (e.g., from DELTA) to directly estimate camera poses and refine world-centric 3D trajectories?

**Conclusion**:

Although the paper addresses an interesting and valuable task, with a well-motivated pipeline and strong performance across several benchmarks, I find the technical novelty to be somewhat limited. The use of decomposed static and dynamic components for camera pose estimation, dynamic refinement, tracking-based correspondence for pose optimization, and the associated loss functions are largely based on existing methods. However, I appreciate the thoughtful integration of these components into a unified pipeline that jointly estimates camera poses and world-centric 3D trajectories, which adds practical value to the field. However, there are several issues in the evaluation protocol (see weaknesses above) that need to be addressed for the results to be fully convincing. I would be happy to increase my score if the authors can address these concerns during the rebuttal.

[1] Karaev, Nikita, et al. "CoTracker3: Simpler and better point tracking by pseudo-labelling real videos." arXiv preprint arXiv:2410.11831 (2024).

[2] Ngo, Tuan Duc, et al. "DELTA: Dense Efficient Long-range 3D Tracking for any video." arXiv preprint arXiv:2410.24211 (2024).

[3] Li, Zhengqi, et al. "MegaSaM: Accurate, fast and robust structure and motion from casual dynamic videos." Proceedings of the Computer Vision and Pattern Recognition Conference. 2025.

[4] Yao, David Yifan, Albert J. Zhai, and Shenlong Wang. "Uni4D: Unifying Visual Foundation Models for 4D Modeling from a Single Video." Proceedings of the Computer Vision and Pattern Recognition Conference. 2025.

**Ethical Concerns:**

["NO or VERY MINOR ethics concerns only"]

**Final Justification:**

Given the strong performance and the importance of the task, I find this work impactful despite the pipeline being built on relatively known components. The integration of these components into a pipeline that achieves impressive results is noteworthy. The authors' rebuttal addressed most of my concerns. I therefore raise my score to borderline accept.

**Limitations:**

Yes

**Quality:**

2

**Strengths And Weaknesses:**

**Strengths**:
* The paper is clearly written and easy to follow, with rigorous and well-presented mathematical formulations.
* Each design choice is well-motivated and thoroughly justified, supported by comprehensive ablation studies that validate their contributions.
* The proposed method outperforms prior approaches in dynamic pose estimation, demonstrating its effectiveness in challenging tracking scenarios.

**Weaknesses**:
* Monocular Depth Inconsistency: The method relies on monocular depth estimators, which are known to produce temporally inconsistent depth, even on static objects. Although the proposed residual term for static 3D tracks and the dynamic 3D track optimization can partially address this issue, I am concerned that depth inconsistency may still significantly affect the final results. Prior works like Mega-SAM and Uni-4D explicitly refine video depth for temporal consistency. It would strengthen this paper if the authors also refined monocular depth across time and potentially provided consistent video depth as an additional output.
* Questionable Depth Evaluation Setup: The evaluation protocol for depth accuracy of dense 3D tracks (Sec. 4.2.2) is unclear. The reported depth accuracy depends on both: (1) The quality of the 2D tracking to correctly localize the corresponding pixel at the target frame; (2) the accuracy of the 3D tracking depth itself. Without guaranteeing perfect 2D correspondence, the comparison between the ground-truth depth at the target pixel and the estimated depth can be misleading. The current evaluation does not disentangle these two sources of error, which undermines the validity of this metric.
* Sparse 3D Tracking Evaluation Details: In Sec. 4.2.3, the authors mention randomly selecting video subsets for evaluation but do not explain the selection criteria or whether this ensures a fair comparison to prior work. A more standardized setup, such as using the TAPVid3D mini-split (50 videos per dataset), would strengthen the evaluation. Additionally, it is unclear whether the sparse 3D tracking is evaluated in world or camera coordinates—this should be specified.
* Lack of Runtime Baseline: The reported runtime of 10 minutes per 30-frame video is considerably slower than methods like Mega-SAM or Uni-4D, though those methods do not output world-centric trajectories. A simple baseline would be to combine camera poses and consistent depths from Mega-SAM [3] or Uni-4D [4] with DELTA’s camera-centric 3D tracking (or dense 2d tracking from Cotracker3 + DELTA’s upsampler) and project them into the world frame. It would be helpful to see how this simpler and potentially faster alternative compares in both accuracy and runtime.

---

> ### Author Rebuttal · Authors · 2025-07-31
>
> 1.  **1) Monocular Depth Inconsistency.**
>
>     Our pipeline is robust to inconsistencies in monocular depth. We tested our pipeline using different monocular depth estimation models and observed consistently depth improvement. This is because our framework jointly optimizes camera poses and enforces consistency across depth predictions from different frames through 2D tracking associations. For detailed experimental results, please refer to our response to Reviewer tFqV, Q1.
>
>     **2) Can the Method Produce Consistent Video Depth?**
>
>     Yes. Since our method aims to achieve dense 3D tracking for nearly all pixels, it can naturally generate consistent video depth outputs. This can be achieved through a straightforward post-processing step (interpolation-based propagation).
>
>     To address your concern, we now describe how we can derive temporally consistent **video depth** from our dense 3D tracking results.
>
>     Assume we have obtained $M$ dense 3D tracking points across the video. At each frame $t \in \{1, 2, \dots, T\}$, we denote the set of 3D tracking points that are visible as $\mathcal{M}_t $, and their camera-centric depths as $D_t$. Let $D_t^{\text{raw}}$ be the raw monocular depth prediction for frame $t$.
>
>     We first align the raw depth with the optimized 3D tracking depth using a **scale ratio**. Unlike some global alignment strategies that compute one scale per frame, we compute **a per-point local scale** between the tracking depths and the raw depth.
>
>     For each tracking point $\mathbf{M}_t^i \in \mathcal{M}_t$, its corresponding raw depth value is $D_t^{\text{raw}}(\mathbf{p}_t^i)$, where $\mathbf{p}_t^i$ is the projected 2D location of $\mathbf{M}_t^i$ on the image plane. The **scale ratio** at that point is defined as:
>     $$
>     r_t^i = \frac{D_t(\mathbf{p}_t^i)}{D_t^{\text{raw}}(\mathbf{p}_t^i)}
>     $$
>     Then, for any pixel $\mathbf{p}_t$ in frame $t$, we estimate its final depth $\hat{D}_t(\mathbf{p}_t)$ via **weighted scale propagation** from nearby 3D tracking points.
>
>     Let $\mathcal{N}_t(\mathbf{p}_t) \subset \mathcal{M}_t$ denote the set of its $k$ nearest neighbors in the image plane. For each neighbor $\mathbf{M}_t^j \in \mathcal{N}_t(\mathbf{p}_t)$, we compute a 3D Euclidean distance between $\mathbf{p}_t$ and $\mathbf{M}_t^j$, using their raw-depth-lifted 3D coordinates. That is:
>
>     $$
>     \mathbf{P}_t = D_t^{\text{raw}}(\mathbf{p}_t) \cdot K^{-1} \tilde{\mathbf{p}}_t, \quad
>     \mathbf{Q}_t^j = D_t^{\text{raw}}(\mathbf{p}_t^j) \cdot K^{-1} \tilde{\mathbf{p}}_t^j
>     $$
>
>     $$
>     d_t^j = \| \mathbf{P}_t - \mathbf{Q}_t^j \|_2
>     $$
>
>     where $K$ is the camera intrinsic matrix and $\tilde{\mathbf{p}}$ denotes the homogeneous coordinate of pixel $\mathbf{p}$.
>
>     We define a weight $w_t^j$ for each neighbor as the **inverse distance weight**:
>
>     $$
>     w_t^j = \frac{1}{d_t^j + \epsilon}
>     $$
>
>     where $\epsilon$ is a small constant to avoid division by zero. The weights are then normalized:
>
>     $$
>     \tilde{w}_ t^j = \frac{w_t^j}{\sum_{j=1}^{k} w_t^j}
>     $$
>
>     Using these weights, we compute the interpolated scale ratio $r_{\mathbf{p}_t}$ at pixel $\mathbf{p}_t$:
>
>     $$
>     r_ {\mathbf{p}_ t} = \sum_ {j=1}^{k} w_ t^j \cdot r_t^j
>     $$
>
>     Finally, we compute the aligned depth at pixel \$\mathbf{p}\$ as:
>
>     $$
>     \hat{D}_ t(\mathbf{p}_ t) = r_ {\mathbf{p}_ t} \cdot D_ t^{\text{raw}}(\mathbf{p}_ t)
>     $$
>
>     Through this approach, we effectively propagate the accurate scale information from our depth-consistent 3D tracks to the entire image, yielding a temporally consistent and spatially coherent video depth. Quantitative results for our video depth can be found in the following table.
>
>     | Category|Method|Sintel|Bonn|TUM D|
>     |-|-|-|-|-|
>     |||Abs Rel ↓ / δ<1.25 ↑|Abs Rel ↓ / δ<1.25 ↑|Abs Rel ↓ / δ<1.25 ↑|
>     ||Depth Anything V2| 0.348 / 59.2 | 0.106 / 92.1 | 0.211 / 78.0 |
>     | Single-frame | Depth Pro| 0.418 / 55.9 | 0.068 / **97.4** | 0.126 / 89.3 |
>     || ZoeDepth| 0.467 / 47.3 | 0.087 / 94.8 | 0.176 / 74.5 |
>     | Video depth| ChronoDepth| 0.687 / 48.6 | 0.100 / 91.1 | - / - |
>     || DepthCrafter| 0.292 / 69.7 | 0.075 / 97.1 | - / -|
>     || Align3R (Depth Pro)| 0.263 / 64.1 | 0.058 / 97.1 | 0.111 / 88.9 |
>     | Joint video depth & pose| Uni4D| 0.236 / 70.6 | **0.057** / 97.2 | 0.091 / 91.5 |
>     || **Ours (DELTA)**| **0.222 / 72.6** | 0.058 / 97.3 | **0.086 / 92.3** |
>
> ---
> 2. **Questionable Depth Evaluation Setup.**
>
>     We agree that our evaluation metric inherently couples depth and correspondence. The reason is that since our goal is to simultaneously reconstruct in 3D and track pixels in the reconstructed 3D space, such a metric evaluates both aspects about 3D reconstructed points and the correspondences across frames.
>
>     To further show the evaluation of the video depth only, we have provided a quantitative comparison of video depth outputs in our response to Q1.
>
>     This evaluation metric only shows the correspondences across frames and depth map. To demonstrate the quality of both camera-centric and world-centric 3D tracking, we also report the sparse camera-centric 3D tracking results in Table 3 of the main text, and the sparse world-centric 3D tracking results in our response to Reviewer wHpN’s first question. Unfortunately, due to the lack of publicly available real-world datasets with ground-truth dense world-centric 3D tracks, we are currently unable to report dense-level evaluation in this setting but only provide the results on world-centric sparse 3D tracks.
>
>
> ---
> 3. **Sparse 3D Tracking Evaluation Details.**
>
>     Thank you for your suggestions. The video subsets for evaluation are selected at fixed intervals: for ADT, we sample one video every 100 videos, and for PStudio, one video every 20 videos. All baseline results reported in the table were reproduced by us using the official repositories under consistent settings. To ensure a more standardized evaluation, we now conduct experiments on the TAPVid3D mini-split.
>
>     | Method| AJ ↑| $\mathrm{APD_{3D}}$ ↑ | OA ↑| AJ ↑| $\mathrm{APD_{3D}}$ ↑ | OA ↑    |
>     |-|-|-|-|-|-|-|
>     | CoTrackerV3 + Uni| 13.9 | 21.0| 88.7    | 14.3    | 23.1| **87.5**|
>     | DELTA| 15.1| 23.2| 89.8    | 14.9    | 24.8| 75.4 |
>     | Ours (CoTrackerV3)   | 22.7    | 31.2| 88.8    | 14.0   | 24.2| **87.5**|
>     | Ours (DELTA) | **23.1**| **32.5**| 90.3    | **15.3**| **25.3**| 75.9    |
>
>     Additionally, the sparse 3D tracking results shown in the above table and in Table 3 of the main text are evaluated in camera coordinates. For world coordinates, we have also provided supplementary results in our response to Reviewer wHpN’s first question.
>
>
>
> ---
> 4. **Lack of Runtime Baseline.**
>
>     Thank you for your suggestions. Due to time constraints, we have currently evaluated one of the baselines you recommended: using camera poses and consistent depths from Uni-4D combined with dense camera-centric 3D tracking from DELTA.
>
>     Since our method and the baseline share the same dense camera-centric 3D tracking pipeline from DELTA, we focus the comparison on the optimization stage and the associated runtime.
>
>     Uni-4D estimates poses in a streaming fashion, i.e., it estimates the pose between frame 1 and 2, then frame 2 and 3, and so forth. Consequently, the total pose estimation time grows linearly with the number of video frames. In contrast, our method estimates poses in a clip-to-global parallel manner, significantly reducing computation time.
>
>     Regarding 3D tracking accuracy, our method achieves better performance by more effectively distinguishing between static and dynamic regions using dynamic masks, and by jointly optimizing both pose and point trajectories. This leads to higher-quality reconstructions compared to the baseline.
>
>     Next, we report detailed results on camera pose estimation (Sintel: frames 30–50) and world-coordinate 3D point tracking (ADT: first 64 frames). The ADT evaluation follows the same experimental setup as described in our response to Reviewer wHpN’s first question. All experiments are conducted on the same CPU and GPU device to ensure fair comparison.
>
>     | Setting | Sintel ATE ↓ | Sintel RTE ↓ | Sintel RPE ↓ | Sintel Avg. Time (min) ↓ | ADT $\mathrm{APD_{3D}}$ ↑ | ADT Avg. Time (min) ↓ |
>     |-|-|-|-|-|-|-|
>     | Baseline (Uni-4D + DELTA) | 0.118| 0.048| 0.610| 19| 68.95|28 |
>     | Ours (DELTA)| **0.087** | **0.036**  | **0.406**| **15**| **75.18** | **20** |
> ---
> 5. **Global pose from clips.**
>
>     We estimate the camera poses for the entire video sequence in a globally consistent manner using the formulation in Equation (2) of the main text. Specifically, we compute the relative poses between adjacent frames within each clip in parallel. Additionally, we estimate the relative pose between the last frame of the i-th clip and the first frame of the (i+1)-th clip. These inter-clip constraints allow us to link all frames together, forming a unified and globally consistent pose trajectory across the entire video.
> ---
> 6. **Alternatively leverage dense 3D tracks.**
>
>     Yes. And we have explored this alternative approach. Specifically, we replaced the dense 2D tracks and monocular depth with dense 3D tracks (obtained from DELTA) for direct camera pose estimation. However, we observed no significant improvement in accuracy, and thus we did not include the results in the main text. For completeness, we report the comparison below:
>
>     | Setting|Sintel ATE ↓| Sintel RTE ↓| Sintel RPE ↓|
>     |-|-|-|-|
>     | Dense 2D tracks + monocular depth | **0.087** | 0.036 | 0.406 |
>     | Dense 3D tracks| 0.092     | **0.035**     | **0.403**     |

---

> > ### Comment · Reviewer_yGCi · 2025-08-03
> >
> > Thank you to the authors for the detailed rebuttal. The additional experiments and clarifications have addressed most of my initial concerns, particularly the monocular depth inconsistency, the ability to obtain consistent video depth. I appreciate the inclusion of runtime and baseline comparisons. However, I still believe Table 2 is misleading due to its conflation of tracking and depth accuracy, and I suggest removing it.
> >
> > Given these improvements, I raise my score to a borderline accept.

---

> > > ### Author Response · Authors · 2025-08-04
> > >
> > > We sincerely thank the reviewer for the thoughtful feedback and for raising the score. We are glad that our additional experiments and clarifications have addressed most of your concerns. We will carefully follow your suggestions and incorporate the improvements into the final version.

---

> ### Author Response · Authors · 2025-08-03
>
> We sincerely appreciate the time and effort you have dedicated to reviewing our manuscript. We have tried our best to provide responses to all of your questions and we hope our responses would address your concerns.
>
> We would be grateful for any final feedback you may have. Please don't hesitate to let us know if further clarifications would be helpful - we remain ready to provide additional details as needed.
>
> Thank you again for your valuable insights and constructive feedback throughout this process.

---

### Official Review · Reviewer_iMow · 2025-07-03

**Clarity:** 3
**Significance:** 3
**Originality:** 3
**Rating:** 4
**Confidence:** 3

**Summary:**

This paper address the problem of extracting dense 3D point tracking from monocular videos in a world-centric 3D coordinate.  It use pretrained visual foundation models to extract sparse (later upsampled to dense) 2D pixel tracks( CoTracker or DELTA for), monocular depth(Unidepth) and dynamics masks (VLM followed by Grounding-Sam). Then it performs stages of optimization to minimize 2D projection loss and some regularization loss as ARAP to get the camera pose, and dense 3D tracks. The author report evaluation metrics like depth accuracy,3D point tracking accuracy  and optical flow accuracy, showing state-of-the-art results compared with both feedforward methods and optimization-based methods.

**Questions:**

Which part is more fragile. This paper proposed an multi-stage optimization that chains the predictions of multiple foundation model together. It would be great to analyze common failure cases, e.g. which part of the stage is less reliable under some cases. I could imagein the part of dynamic mask, since it's generated with VLM and grounded SAM, it might miss some objects with semantic feature of static objects but are moving in the video.


Novel view visualization will be better.  In the supp, I found videos of lifted 3D dynamic pointclouds at a static viewpoint. It would be very beneficial to include pointclouds visualizations of novel views, since novel views give better visualization of the 3D geometry.

**Ethical Concerns:**

["NO or VERY MINOR ethics concerns only"]

**Final Justification:**

I read the author's rebuttal, and I am satisfied with the author's analysis on the potential failure mode in his multi-stage algorithm and the promise to add novel view visualizations of the reconstruction. I raise my score to 4: weak accept.

**Limitations:**

Yes.

**Quality:**

3

**Strengths And Weaknesses:**

Strength:

This paper tackles a very important problem, 3D dynamic tracking in a world-coordinates, and deliveres good results, especially quantitative results.  The writting is also in general clear.



Weakness:

1. I think the inherent weakness is about the complexity of the pipeline, making it hard to understand the current bottleneck and failure mode. I encourage the author providing more insight to it (mentioned in the question section as well)
2. Another inherent weakness is that the proposed method is not a end-to-end feedforward model. It's an adhoc optimization methods, which makes it hard to scale (if we have more data, the performance of the optimization based methods might grow slower than end-to-end methods)

---

> ### Author Rebuttal · Authors · 2025-07-31
>
> 1. **More insight to the method.**
>
>     Our proposed method follows a multi-stage pipeline consisting of four key components:
>
>     1. **All-frame Dense 2D Tracking**
>     2. **Initial Camera Pose Estimation** using monocular depth and static 2D tracks
>     3. **Joint Refinement** of camera poses and static 3D tracks
>     4. **Estimation of Dynamic 3D Tracks**
>
>     Throughout this process, we rely on off-the-shelf 2D tracking models to generate dense 2D tracks, which serve as the foundation of our pipeline. Monocular depth provides critical geometric priors that help prevent the optimization from getting trapped in poor local minima. We have validated the robustness of both components—monocular depth and 2D tracking—under commonly encountered conditions, as discussed in our response to Reviewer wHpN (Q3).
>
>     We also leverage dynamic masks to distinguish between static and dynamic regions. However, due to the imperfect nature of existing dynamic segmentation techniques, we design specific mechanisms to mitigate their limitations:
>
>     1. **Static masks may include dynamic objects.**
>     To address this, we introduce an *as-static-as-possible* constraint, which encourages only truly static regions to be used during pose optimization. This design is evaluated in our ablation study (see Figure 4 in the main text and our response to Reviewer wHpN, Q2).
>
>     2. **Dynamic masks may include static objects.**
>     As discussed in our response to Reviewer wHpN (Q7), this does not significantly impact the accuracy of camera pose estimation. Our method primarily relies on the consistency of truly static regions to recover camera poses, so a small number of misclassified static areas in dynamic masks has minimal effect. Furthermore, during the dynamic 3D tracking stage, we apply an additional optimization step on the dynamic regions. This step enforces 2D track consistency, which can correct some of the earlier misclassifications and ensures accurate 3D tracking.
>
>     In summary, our pipeline demonstrates good robustness to variations in monocular depth, 2D tracks, and dynamic masks. However, it may fail under the following two challenging scenarios:
>     1. **When dynamic objects dominate the camera’s field of view**, camera pose estimation becomes unreliable. If the entire scene is dynamic, even a human observer would struggle to distinguish motion, making reliable pose recovery infeasible.
>
>     2. **When the scene lacks texture**, the quality of 2D tracks becomes poor. Current state-of-the-art 2D tracking models still struggle to generalize to textureless environments, which remains a critical direction for future research.
>
>
> ---
> 2. **Justification of Optimization-based Framework.**
>
>     We appreciate the reviewer’s comment and agree that end2end methods are more attractive and promising due to their effiency. However, we believe such methods offer distinct advantages, particularly for the task of *world-centric dense 3D tracking*, where obtaining large-scale annotated data in real-world scenarios is extremely difficult.
>
>     Unlike end-to-end approaches that often require substantial supervision—such as accurate 3D annotations and ground-truth camera poses—our method decomposes the problem into several more tractable and annotation-efficient sub-tasks, including monocular depth estimation, dynamic region segmentation, and 2D tracking. These individual tasks have demonstrated strong robustness with recent advancements. Our proposed optimization framework then integrates them in a coherent and geometrically consistent manner, enabling accurate and scalable world-centric dense 3D tracking without relying on fully annotated datasets.
>
>     Furthermore, we emphasize that end-to-end and optimization-based methods are not mutually exclusive but rather complementary. For instance, St4RTrack leverages large-scale synthetic datasets for training, while still incorporating optimization steps to improve generalization to real-world data. Similarly, VGGT benefits from applying bundle adjustment, resulting in improved reconstruction accuracy (see Table 1 in VGGT). These examples highlight the practical utility of optimization-based techniques—especially when combined with end-to-end models—to enhance performance in complex and unconstrained real-world settings.
> ---
> 3. **Novel view visualization.**
>
>     Thank you for your suggestions. We will update the videos in the revised version to include visualizations from more diverse viewpoints and also provide interactive point cloud visualizations for clearer result interpretation. However, due to this year’s policy, we are unable to include these updated results during the rebuttal phase. We apologize for the inconvenience.

---

> > ### Comment · Reviewer_iMow · 2025-08-05
> >
> > I appreciate the author's efforts on providing more understanding of the pipelines.
> >
> > I am kind of satisfied by the author's promise on providing novel view visualizations of the reconstruction.
> >
> > I would consider raising my score.

---

> > > ### Author Response · Authors · 2025-08-06
> > >
> > > We sincerely appreciate your thoughtful comments and your consideration to raise the score. We're glad the additional clarifications have been helpful in conveying the pipeline more clearly.  We will carefully follow your suggestions and incorporate the improvements into the final version. Your feedback is very instrumental in improving the quality and clarity of our work.

---

> > > ### Author Response · Authors · 2025-08-09
> > >
> > > Dear Reviewer iMow,
> > >
> > > We sincerely thank you for your thoughtful feedback and engaging discussion. As the rebuttal period draws to a close, we would like to gently note that there is still a brief window should you wish to make any final refinements to your review, as well as for submitting your final rating and acknowledgment. We truly value the time, effort, and insightful contributions you have so generously shared to help strengthen our work. Please rest assured that we will carefully consider your suggestions and thoughtfully incorporate the revised content into the final version.
> > >
> > > Sincerely,
> > >
> > > Authors

---

> ### Author Response · Authors · 2025-08-03
>
> We sincerely appreciate the time and effort you have dedicated to reviewing our manuscript. We have tried our best to provide responses to all of your questions and we hope our responses would address your concerns.
>
> We would be grateful for any final feedback you may have. Please don't hesitate to let us know if further clarifications would be helpful - we remain ready to provide additional details as needed.
>
> Thank you again for your valuable insights and constructive feedback throughout this process.

---

### Official Review · Reviewer_wHpN · 2025-07-03

**Clarity:** 4
**Significance:** 4
**Originality:** 4
**Rating:** 5
**Confidence:** 4

**Summary:**

This paper proposes an optimization-based method to estimate camera poses and 3D dense tracking of dynamic objects in the world-coordinate system from monocular RGB inputs. This is achieved mainly by building on top of several vision foundation models such as UniDepth for depth prior, DELTA for 2D tracks, Uni4D for dynamic object masks. The sparse 2D tracks are lifted to dense 2D tracks using DELTA uplifting module. Then, tracking is performed on every frame and the overlapping tracks are removed. After the final optimization, the method achieves new state-of-the-art performance on several benchmarks, including camera pose estimation, depth accuracy, and 3D tracking performance.

**Questions:**

* It is mentioned that "almost all" means that the method filters some noisy and outlier tracks. However, I haven't found information about how this filtering is done. How much it affects accuracy? Is the threshold selection robust?
* It is also mentioned (P3L120) that VGGT and other methods enable full 4D reconstruction. However, this is not true as VGGT works only for static scenes. Could you please elaborate?
* L192 mentions that dynamic object masks are not accurate, and some dynamic objects can still exist in "static" regions. Can the error come also from dynamic object masks containing some actually static regions? Can this be fixed by the optimization?

**Ethical Concerns:**

["NO or VERY MINOR ethics concerns only"]

**Final Justification:**

The detailed rebuttal addressed most of my concerns are addressed, but I still do not agree that VGGT can be used for 4D reconstruction since it even outputs a static point cloud. It's also very important to add those additional results in the camera ready. I keep my positive rating.

**Limitations:**

Yes, the limitations are adequately discussed in the supplementary materials.

**Paper Formatting Concerns:**

No concerns.

**Quality:**

3

**Strengths And Weaknesses:**

Strengths:
* This is the first method to estimate camera poses and 3D dynamic point tracking in world-coordinate system from monocular RGB streams. This is an important task, and the proposed solution will benefit many other downstream tasks.
* A wide variety of tasks are also solved by the proposed method (camera pose, depth, 3D tracking), and new state-of-the-art results are achieved on several datasets.
* The method is implemented in such a way that it runs in some reasonable time, by means such as camera pose estimation at a clip level.
* The paper is well-written and seems to be straightforward to reproduce.


Weaknesses:
* St4RTrack is mentioned and it seems to be addressing a similar setup. Since their code is not released yet, would it be possible to follow some of their evaluation protocols to see how the proposed method compares to St4RTrack?
* Dynamic masks are essentially also optimized by allowing other parts of the scene become dynamic. Why is the accuracy of those refined masks not evaluated?
* It is mentioned that the monocular priors are not required to be accurate. However, this is not so well evaluated. What happens when there are inaccuracies in the depth predictions? What if other depth predictions are used? L139-147 are confusing because they mention that CoTracker or DELTA can be used (also, GroundingSAM or SegmentAnyMotion can be used, etc). However, this is not ablated. If I understood it correctly, only one type of each monocular prior was tried. It is important to ablate this as well.
* As mentioned on L284, the performance boost is mainly visible for dynamic camera scenes. When the camera is static, the performance improvement is quite small.

---

> ### Author Rebuttal · Authors · 2025-07-31
>
> 1. **Follow St4RTrack evaluation protocols.**
>
>     Thank you for pointing this out. We agree that comparing agains St4RTrack under similar evaluation protocols is valuable. While the official code and data split for St4RTrack are not publicly available, we made our best effort to follow their described protocol for World Coordinate 3D Point Tracking.
>
>     St4RTrack states:
>
>     > "Our benchmark comprises four datasets, each containing **50 sequences** of **64 frames**. We then compute the prediction error and measure the percentage of points whose error falls below a threshold $\delta_{\mathrm{3D}}$ ( with **$\delta_{\mathrm{3D}} \in \{0.1\text{m}, 0.3\text{m}, 0.5\text{m}, 1.0\text{m}\}$**) over **the first 64 frames**. From these, we **randomly sample 50 sequences** per dataset for evaluation."
>
>     Since the official code and exact data split of St4RTrack are not publicly available, we adopted the **TAPVid3D mini-split** (50 videos per dataset officially provided by TAPVid3D) and followed their described protocol by evaluating on the **first 64 frames** of each sequence.
>
>     While we cannot guarantee an identical evaluation setting, and therefore cannot make a strictly fair comparison with St4RTrack, we aimed to provide as meaningful a reference as possible. To this end, we re-implemented the key baselines reported in St4RTrack—**SpaTracker+MonST3R**—and evaluated them using the same **$\mathrm{APD_{3D}}$** metric, after **global median alignment**, as specified in their paper.
>
>     | Method             | ADT ↑     | PStudio ↑ |
>     | ------------------ | --------- | --------- |
>     | SpaTracker+MonST3R | 54.15     | 63.10\*   |
>     | **Ours (DELTA)**   | **75.18** | **65.23** |
>     \* *Note: PStudio does not include camera motion; therefore, we only applied SpaTracker for this evaluation.*
>
>     As shown, our method outperforms the key reported baseline. We believe this result provides strong evidence for the effectiveness of our approach under comparable conditions, despite the absence of results from St4RTrack.
>
>
> ---
>
> 2. **The accuracy of dynamic masks.**
>
>     Thank you for the insightful comment. Following your suggestion, we evaluated our method on the **DAVIS2016-Moving** benchmark using the evaluation protocol introduced in **Segment Any Motion** \[1], which reports region similarity **J** and contour accuracy **F**.
>
>     | Method                      | J ↑      | F ↑      |
>     | --------------------------- | -------- | -------- |
>     | Baseline (VLM + GroundingSAM)   | 87.6     | 88.0     |
>     | Baseline + Ours (Refined Masks) | **91.8** | **91.2** |
>
>     Our refinement strategy improves both J and F scores, demonstrating that the optimized dynamic masks can yield more accurate segmentations.
>
>     \[1] Huang, Nan, et al. *Segment Any Motion in Videos*. CVPR 2025.
>
>
> ---
>
> 3. **Ablation on Each Component.**
>
>     Thank you very much for your suggestions. Since the ablations for different trackers (e.g., **CoTrackerV3** and **DELTA**) have already been provided in Tables 1–4 in the main text, we focus here on the following two components:
>
>     1. Different **monocular depth estimation models**
>     2. Different **dynamic mask segmentation methods**
>
>     For all experiments, we fixed the tracking component to **DELTA** and evaluated both the **camera pose estimation accuracy** and the **depth accuracy** of the dense 3D tracks on the **Sintel** dataset.
>
>     ---
>
>     **Ablation on Monocular Depth Estimation Models**
>
>     | Method           | ATE ↓     | RTE ↓     | RPE ↓     | Abs Rel ↓ | δ < 1.25 ↑ |
>     | ---------------- | --------- | --------- | --------- | --------- | ---------- |
>     | ZoeDepth         | —         | —         | —         | 0.814     | 46.1       |
>     | Depth Pro        | —         | —         | —         | 0.813     | 50.7       |
>     | UniDepth         | —         | —         | —         | 0.636     | 63.1       |
>     | Ours (ZoeDepth)  | 0.093     | 0.038     | 0.418     | 0.236     | 72.1       |
>     | Ours (Depth Pro) | 0.101     | 0.036     | 0.434     | 0.228     | 72.6       |
>     | Ours (UniDepth)  | **0.088** | **0.035** | **0.410** | **0.218** | **73.3**   |
>
>     As shown above, our method consistently improves upon the raw depth predictions across all monocular depth estimation models，indicating that our pipeline is robust to the choice of depth backbone—even when the original predictions are less accurate.
>
>     ---
>
>     **Ablation on Dynamic Mask Segmentators**
>
>     | Method                    | ATE ↓     | RTE ↓     | RPE ↓     | Abs Rel ↓ | δ < 1.25 ↑ |
>     | ------------------------- | --------- | --------- | --------- | --------- | ---------- |
>     | Ours + VLM + GroundingSAM | **0.088** | **0.035** | 0.410 | **0.218** | **73.3**   |
>     | Ours + Segment Any Motion | 0.093     | 0.041     | **0.379**     | 0.224     | **73.3**    |
>
>     We also evaluate different sources of dynamic mask segmentations and observe comparable performance, further demonstrating the robustness of our pipeline.
>
> ---
>
> 4. **When the camera is static, the performance improvement is quite small.**
>
>     Our method is specifically optimized to leverage geometric consistency from stereo cues, which are more effective under dynamic camera motions. In static camera scenes, the benefits are naturally less pronounced.
> ---
> 5. **Filtering some noisy and outlier tracks.**
>
>
>     The filtering process is performed during the “Tracking every frame” stage (Lines 162–167). Specifically, as stated in the paper:
>
>     > *“To avoid wasting computation on these redundant 2D tracks in the subsequent computation, if a pixel resides near the tracking trajectory of arbitrary visible previous 2D tracks, then we discard the pixel.”* (Lines 165–167)
>
>     To implement this, we first compute the **complement set** by discarding pixels that lie in any previously visible 2D track trajectory. However, this operation may introduce **isolated pixels**, which are typically not meaningful for downstream scene understanding. To mitigate this, we construct **connected components** from the newly selected 2D tracking points and apply a size threshold $\tau$ to remove small components. Only connected regions with more than $\tau$ pixels are retained. This ensures that the preserved 2D tracks are **geometrically meaningful** and more likely to correspond to **coherent object parts**.
>
>     We found that this filtering procedure consistently improves accuracy, as most filtered points are either outliers or redundantly close to already tracked regions. Moreover, the threshold $\tau$ is robust across different scenes: we use a fixed value of $\tau = 40$ in all experiments without additional tuning.
>
>     **Ablation on $\tau$ on Sintel and Bonn datasets:**
>     | Setting      | **Sintel** |           |           | **Bonn**  |           |           |
>     | ------------ | ---------- | --------- | --------- | --------- | --------- | --------- |
>     |              | ATE ↓      | RTE ↓     | RPE ↓     | ATE ↓     | RTE ↓     | RPE ↓     |
>     | w/o $\tau$ | 0.105      | 0.038     | 0.442     | 0.018     | 0.007     | 0.601     |
>     | w/ $\tau$  | **0.088**  | **0.035** | **0.410** | **0.016** | **0.005** | **0.564** |
>
>     The results show that the filtering mechanism leads to consistent improvements across different datasets.
>
> ---
> 6. **Clarification on VGGT’s Capability for 4D Reconstruction in Dynamic Scenes.**
>
>     Thank you for pointing this out. While VGGT was originally designed for static scenes, we found that it was trained on a variety of datasets, including those with dynamic content such as PointOdyssey and Virtual KITTI, as mentioned in their paper. Furthermore, recent works like Monst3r and Align3r have demonstrated that the 3R framework can be fine-tuned directly on dynamic datasets to achieve competitive dynamic reconstruction performance. In our own experiments, we also observed that VGGT is able to reconstruct dynamic scenes reasonably well in some cases. Based on this evidence, we therefore used the term "full 4D reconstruction" in a broad sense to indicate the potential capability of such methods to handle dynamic content. That said, we agree that VGGT is not explicitly designed for dynamic scenes, and its performance on such data may vary depending on scene complexity and motion types.
> ---
> 7. **Dynamic object masks containing some actually static regions.**
>
>     Dynamic object masks may indeed contain some regions that are actually static. However, this has minimal impact on the accuracy of the camera pose estimation because our method primarily relies on the consistency of truly static regions to solve for precise camera poses. Therefore, including a few static areas within the dynamic masks does not degrade the camera pose results.
>
>     Furthermore, for 3D tracking, we perform an additional optimization step focused on the dynamic regions. This optimization  can leverage the consistency constraints of 2D tracks to effectively reconstruct misclassified static regions, ensuring that the final 3D tracking accuracy is preserved.

---

> > ### Comment · Reviewer_wHpN · 2025-08-05
> >
> > Thank you for your detailed rebuttal. Some of my concerns are addressed, but I still do not agree that VGGT can be used for 4D reconstruction since it even outputs a static point cloud. It's also very important to add those additional results in the camera ready. I keep my positive rating.

---

> > > ### Author Response · Authors · 2025-08-05
> > >
> > > Thank you for the follow-up. We agree with your assessment regarding the limitations of VGGT for 4D reconstruction. You are absolutely right that VGGT outputs a static point cloud and was not originally designed to model temporal dynamics explicitly. As such, it should not be considered a dedicated method for 4D reconstruction. We will revise the wording in the final version to avoid any potential misinterpretation and will include the additional results and clarifications you recommended in the camera-ready version.
> > >
> > > We sincerely appreciate your constructive feedback and your positive rating.

---

> ### Author Response · Authors · 2025-08-03
>
> We sincerely appreciate the time and effort you have dedicated to reviewing our manuscript. We have tried our best to provide responses to all of your questions and we hope our responses would address your concerns.
>
> We would be grateful for any final feedback you may have. Please don't hesitate to let us know if further clarifications would be helpful - we remain ready to provide additional details as needed.
>
> Thank you again for your valuable insights and constructive feedback throughout this process.

---

### Official Review · Reviewer_tFqV · 2025-07-09

**Clarity:** 3
**Significance:** 2
**Originality:** 3
**Rating:** 5
**Confidence:** 2

**Summary:**

This paper proposes a dense 3d tracking method for monocular vidoes, named TrackingWorld. Given a video consisting of T frames, TrackingWorld will output 3D trajectories which are defined in world-centric 3D space for almost all pixels. Firstly, TrackingWorld adopt a preprocess procedure which output 2D sparse tracking results, dynamic masks and depth maps for all frames. Then, a upsampler module is used to output dense 2D tracks. After obtaining dense 2D tracks, points in static regions are used to estimate the initial camera pose and a dynamic background refinement module is used for accurate camera estimation. Finally, camera pose and depth are used to obtain 3D tracks of dynamic regions with depth consistency, as-rigid-as-possible and temporal smoothness constraints. The authors conduct experiments on different benchmarks including camera pose estimation, depth estimation, sparse 3d tracking and dense 2d tracking tasks. The proposed methods achieve superior performance compared to other SOTA methods.

**Questions:**

1. Please add ablation study to evaluate the robustness of the proposed method with different depth estimation methods.
2. Please explain the necessity of the upsampler module.

**Ethical Concerns:**

["NO or VERY MINOR ethics concerns only"]

**Final Justification:**

The author's rebuttal addressed my concerns. I raise my score to accept.

**Limitations:**

Yes.

**Paper Formatting Concerns:**

No formatting issues.

**Quality:**

2

**Strengths And Weaknesses:**

Strengths
1. This paper proposed a pipeline without training to obtain dense 3d tracking results for a monocular video. The pipeline adopt multiple vison foundation models such as 2d tracking, segmentation, depth estimation and use a optimization-based framework to obtain results.
2. The authors conduct experiments on multiple benchmarks and show the proposed method can achieve superior performances across all metrics.

Weaknesses
1. The proposed method relies on depth estimation models for the 2D-to-3D lifting process heavily. The authors should conduct ablation studies to evaluate the pipeline's robustness to different depth estimation models.
2. What is the necessity of employing an upsampler module for 2D track upsampling? The paper lacks ablation studies to explain.

---

> ### Author Rebuttal · Authors · 2025-07-31
>
> 1. **Ablation Study on Different Depth Estimation Models.**
>
>     Thank you for the insightful comment. We agree that the robustness of our method to different depth estimation backbones is important. To address this concern, we conducted an ablation study using three commonly used monocular depth estimation models: **ZoeDepth \[1]**, **Depth Pro \[2]**, and **UniDepth \[3]**. For all experiments, we fixed the tracking component to **DELTA \[4]** and evaluated both the camera pose estimation accuracy and the depth accuracy of the dense 3D tracks on the **Sintel** dataset.
>
>     | Method           | ATE ↓     | RTE ↓     | RPE ↓     | Abs Rel ↓ | δ < 1.25 ↑ |
>     | ---------------- | --------- | --------- | --------- | --------- | ---------- |
>     | ZoeDepth         | —         | —         | —         | 0.814     | 46.1       |
>     | Depth Pro        | —         | —         | —         | 0.813     | 50.7       |
>     | UniDepth         | —         | —         | —         | 0.636     | 63.1       |
>     | Ours (ZoeDepth)  | 0.093     | 0.038     | 0.418     | 0.236     | 72.1       |
>     | Ours (Depth Pro) | 0.101     | 0.036     | 0.434     | 0.228     | 72.6       |
>     | Ours (UniDepth)  | **0.088** | **0.035** | **0.410** | **0.218** | **73.3**   |
>
>     As shown above, our method consistently improves over raw depth predictions across all depth models, especially in downstream tasks such as camera pose estimation. This demonstrates that our pipeline is robust to different depth estimation backbones. We will include this ablation table in the revised submission.
>
> ---
>
> 2. **Necessity of the 2D upsampler module and missing ablation.**
>
>     Thank you for the question. The 2D upsampler is crucial for achieving **efficient dense tracking**. Directly predicting dense 2D correspondences (e.g., using CoTrackerV3 \[5]) is computationally expensive and memory-intensive, with no clear accuracy gain. To validate this, we compare CoTrackerV3 with and without our upsampler on the CVO-Clean dataset (7-frame sequences). All experiments are run on an A6000 GPU to avoid OOM issues.
>
>     | Method           | EPE ↓    | IoU ↑    | Avg. Time (min) ↓ |
>     | ---------------- | -------- | -------- | ----------------- |
>     | CoTrackerV3      | 1.45     | 76.8     | 3.00              |
>     | CoTrackerV3 + Up | **1.24** | **80.9** | **0.25**          |
>
>     As shown, the upsampler accuracy (↓EPE, ↑IoU) and drastically reduces runtime (\~12× speed-up). This supports our design choice. Additional evidence can also be found in Table 2 and Table 3 of the main text of the DELTA paper.
>
> [1] Bhat, S. F., Birkl, R., Wofk, D., et al. *Zoedepth: Zero-shot Transfer by Combining Relative and Metric Depth*. arXiv 2023.
>
> [2] Bochkovskii, A., Delaunoy, A., Germain, H., et al. *Depth Pro: Sharp Monocular Metric Depth in Less Than a Second*. ICLR 2025.
>
> [3] Piccinelli, L., Yang, Y. H., Sakaridis, C., et al. *UniDepth: Universal Monocular Metric Depth Estimation*. CVPR 2024.
>
> [4] Ngo, T. D., Zhuang, P., Gan, C., et al. *DELTA: Dense Efficient Long-Range 3D Tracking for Any Video*. ICLR 2025.
>
> [5] Karaev, N., Makarov, I., Wang, J., et al. *CoTracker3: Simpler and Better Point Tracking by Pseudo-Labelling Real Videos*. ICCV 2025.

---

> > ### Comment · Reviewer_tFqV · 2025-08-04
> >
> > The rebuttal addressed my concerns. I keep my rating as positive.

---

> > > ### Author Response · Authors · 2025-08-04
> > >
> > > Thank you very much for your feedback. We are glad that our rebuttal effectively addressed your concerns.  We will carefully follow your suggestions and incorporate the improvements into the final version.

---

> ### Author Response · Authors · 2025-08-03
>
> We sincerely appreciate the time and effort you have dedicated to reviewing our manuscript. We have tried our best to provide responses to all of your questions and we hope our responses would address your concerns.
>
> We would be grateful for any final feedback you may have. Please don't hesitate to let us know if further clarifications would be helpful - we remain ready to provide additional details as needed.
>
> Thank you again for your valuable insights and constructive feedback throughout this process.

---

### Decision · Program_Chairs · 2025-09-17

**Decision:**

Accept (poster)

**Comment:**

The paper presents TrackingWorld, a optimization‑based pipeline that jointly estimates camera poses and dense world‑centric 3D trajectories from a monocular video. The authors convincingly integrate several high‑performing vision foundation models (depth, 2D tracks, dynamic masks) and demonstrate consistent, state‑of‑the‑art results on multiple benchmarks.
The authors have successfully addressed most of the reviewers' concerns (depth‑model robustness, necessity of the upsampler, filtering of noisy tracks, etc.), resulting in two "accept" and two "borderline accept" ratings for the paper. Given the solid technical contribution, comprehensive evaluation, and satisfactory responses to all major reviewer questions, we recommend accept the paper.